# The mechanism of tidal triggering of earthquakes at mid-ocean ridges

Christopher H. Scholz [1,3], Yen Joe Tan [1,3] & Fabien Albino [2]

The strong tidal triggering of mid-ocean ridge earthquakes has remained unexplained because the earthquakes occur preferentially during low tide, when normal faulting earthquakes should be inhibited. Using Axial Volcano on the Juan de Fuca ridge as an example, we show that the axial magma chamber inflates/deflates in response to tidal stresses, producing Coulomb stresses on the faults that are opposite in sign to those produced by the tides. When the magma chamber's bulk modulus is sufficiently low, the phase of tidal triggering is inverted. We find that the stress dependence of seismicity rate conforms to triggering theory over the entire tidal stress range. There is no triggering stress threshold and stress shadowing is just a continuous function of stress decrease. We find the viscous friction parameter $A$ to be an order of magnitude smaller than laboratory measurements. The high tidal sensitivity at Axial Volcano results from the shallow earthquake depths.

[1] Lamont-Doherty Earth Observatory, Columbia University, Palisades, NY 10964, USA. [2] School of Earth Sciences, University of Bristol, Bristol B58 1RJ, UK. [3] These authors contributed equally: Christopher H. Scholz, Yen Joe Tan. Correspondence and requests for materials should be addressed to C.H.S. (email: scholz@ldeo.columbia.edu) or to Y.J.T. (email: yjt@ldeo.columbia.edu)

Tidal triggering of earthquakes has been found to be elusive despite a long search[1–3]. On the continents, the signal is so weak that a significant statistical correlation between earthquakes and tides can be detected only with very large catalogues[4]. In the oceans, where loading from ocean tides result in tidal stresses an order of magnitude larger than the solid earth tides, the tidal triggering signal has been found to be stronger. Cochran et al.[5] used a large catalog of oceanic earthquakes to show that shallow thrust earthquakes may be found to correlate with maximum tidally generated Coulomb stresses when the tides are large enough. The earthquakes preferentially occur at low water when the normal stresses on low-angle thrust faults are reduced such that they become unclamped. Much stronger tidal triggering has been observed with ocean bottom seismometer networks in magmatic areas at mid-ocean ridges[6–10]. These are the most promising places to test theories of earthquake triggering. In these cases, however, even the most basic mechanism of the triggering is not understood because seismicity peaks at low tides, when it should be inhibited on normal faults. The most well-studied of these cases is at Axial Volcano on the Juan de Fuca ridge. We shall study this case, and later see if the results obtained there can also be applied to the other mid-ocean ridges.

At Axial Volcano, the rate of normal faulting earthquakes is maximum at low tide. We show that they are driven by the tidally induced inflation of the magma chamber. The seismicity rate is modulated by tidal stresses at all tidal phases in agreement with triggering models based on nucleation theory. These results show that the viscous friction parameter $A$ must be much smaller than indicated in laboratory experiments.

## Results

**Tidal triggering of earthquake at Axial Volcano.** Axial Volcano, which is at the intersection of a mid-ocean ridge with a hotspot (Fig. 1), erupts on a decadal time scale. Each eruption is followed by a caldera collapse accompanied by thrusting on outwardly dipping ring faults, followed by a re-inflation period, at the latter stages of which the ring faults become reactivated in normal faulting[11–13]. The best observations of tidal triggering were for the normal faulting earthquakes in the months prior to the 2015 eruption[7].

At Axial Volcano, the ocean tides are very large (3 m) so that ocean loading dominates the solid earth tides and the vertical tidal stress dominates and is in phase with the ocean tides (Supplementary Fig. 1), so we need only consider the vertical

component in our analysis. Tension is taken as positive for tidal stresses, so the maximum tidal stress corresponds to the minimum water depth. To avoid ambiguity, in this paper we will refer to high and low tides in the conventional way as high and low water, recalling that low water produces tension and high water produces compression.

Figure 1 shows a cross-section view of the seismicity for the three months prior to the 2015 eruption, which illuminates the ring faults. This data set contains ~60,000 earthquakes with a magnitude of completeness $(M_c) = 0.1$[14]. Figure 2 shows a histogram of the seismicity plotted as a function of tidal phase, in which 0° is the maximum low tide. The correlation is obvious and requires no statistical treatment. It was first proposed that this was also a case of fault unclamping[6,8,10], but when it was established that these earthquakes were dominated by normal faulting[12] this viewpoint became untenable. Both the seismicity trends in Fig. 1 and the focal mechanisms[12] indicate a mean fault dip of 67°. The reduction of vertical stress brought about by low tide will produce a Coulomb-stress change on such steeply dipping normal faults that inhibits their slip. It is, rather, the high tides that will produce a Coulomb stress on the faults that encourages slip. This seeming paradox is resolved by including the effect of the axial magma chamber on the distribution of stress.

**The response of the magma chamber.** The red curve in Fig. 1 delineates the roof of the axial magma chamber obtained from seismic imaging[15]. Inflation of the magma chamber drives the normal faulting on the ring faults. This is demonstrated in Fig. 3, where we show east–west cross-sections of a 3D model containing a magma chamber with dimensions defined by seismic imaging[15,16]. Figure 3a shows the Coulomb failure stress change, $\Delta CFS = \Delta\tau + \mu\Delta\sigma$, on 67° dipping faults that results from a magma chamber overpressure of 1 MPa ($\Delta\tau$ is the change in shear stress resolved on the fault in the slip direction, $\Delta\sigma$ is the change in normal stress on the fault plane, and $\mu$ is the friction coefficient). Positive (red) $\Delta CFS$ values encourage normal fault slip, negative ones (blue) inhibit it. The primary features in Fig. 3a are the zones of positive $\Delta CFS$ that correspond to the seismicity shown in Fig. 1. See Methods for details about the model.

Because the magma chamber is a soft inclusion, its presence will profoundly affect the stress field in its vicinity resulting from any external load. We simulate the response to tides by

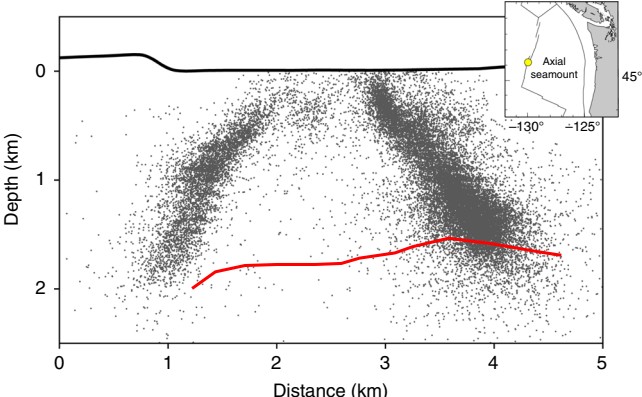

**Fig. 1** Cross-section of seismicity at Axial Volcano. East–west cross-section of seismicity in 3 months preceding the 2015 eruption at Axial Volcano. Red curve is the roof of the axial magma chamber[15]. The bathymetry is from a compilation of latest cruises, the most recent in 2010[62]. Inset, location map for Axial Volcano

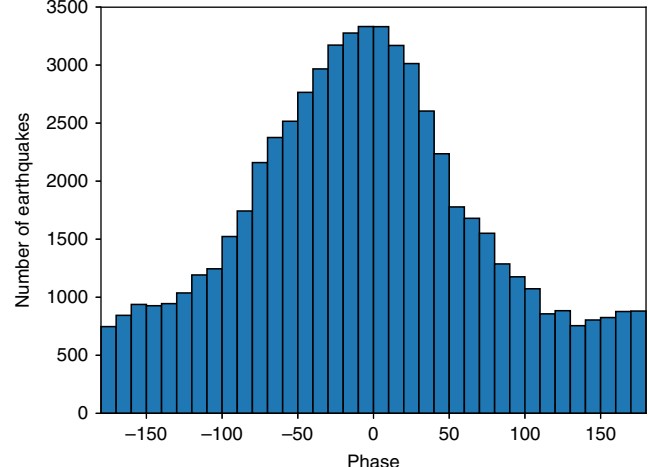

**Fig. 2** Seismicity versus tidal phase. Histogram of earthquakes plotted vs. the phase of the vertical component of the tidal stress, in which 0° is the peak stress (tension is positive), which corresponds to the lowest ocean height

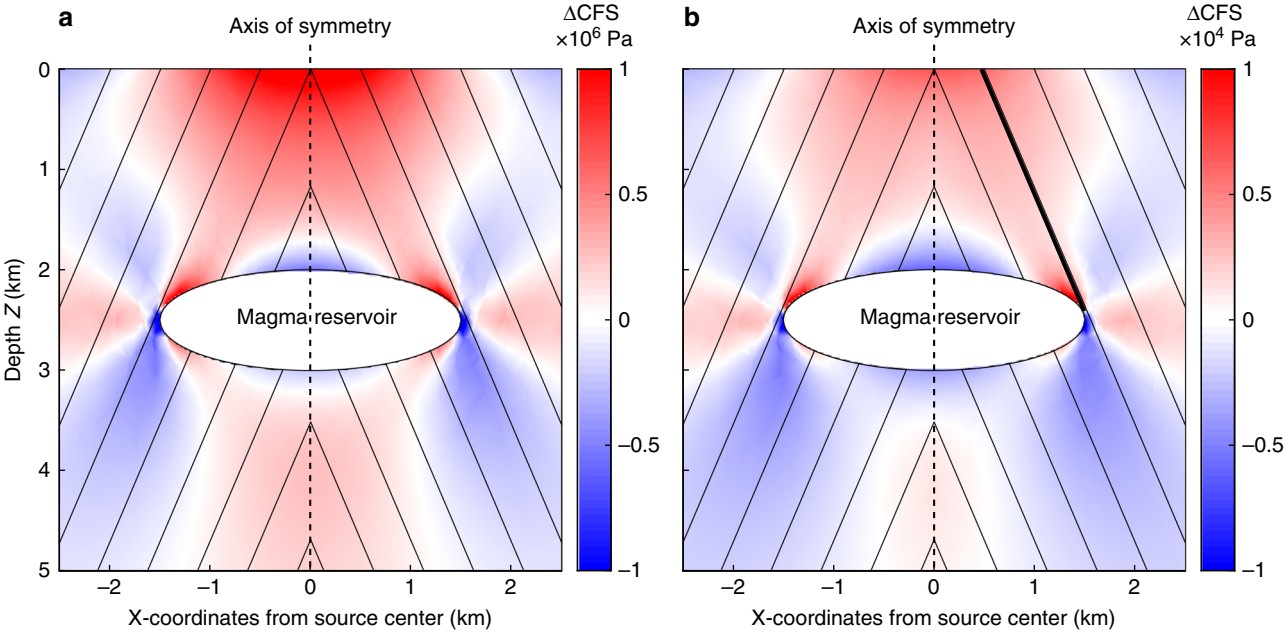

**Fig. 3** Coulomb stress on normal faults above the magma chamber. Distribution of Coulomb stress changes on 67° dipping normal faults near the axial magma chamber. Positive values favor fault slip, negative values inhibit it. **a** For an overpressure of 1 MPa within the magma chamber with friction coefficient of the faults $\mu = 0.8$. **b** For a decrease in vertical stress equivalent to a reduction in water level of 1 m. In **b** an effective friction $\mu' = 0.4$ is used. The bulk modulus of the rock is assumed to be $K_r = 55$ GPa, and the bulk modulus of the magma chamber is assumed to be $K_m = 1$ GPa. The heavy line in Fig. 3b indicates the fault upon which $\Delta$CFS was measured

calculating the distribution of $\Delta$CFS on 67° dipping faults resulting from a reduction in vertical stress corresponding to a 1 m drop in the ocean tide. This is shown in Fig. 3b. The pattern is very similar to that of Fig. 3a, demonstrating how a low tide can stimulate activity on these faults. This pattern arises because the reduction of vertical stress causes the magma chamber, owing to its higher compressibility, to inflate relative to the surrounding rock, which produces a stress field congruent with that of Fig. 3a. This is superimposed on a uniform $\Delta$CFS from the tidal stress, which is negative in the case of a low tide. Likewise, high tides cause the magma chamber to deflate, which also produces Coulomb stresses opposite in sign to the tidal ones. Which component is larger determines whether normal faulting earthquakes are stimulated by the low tide or the high tide.

The relative expansion of the magma chamber depends inversely with $K_m/K_r$, the bulk modulus of the magma chamber relative to that of the surrounding rock, so this is the critical parameter that determines the behavior of the system. In the calculation of Fig. 3a we used $\mu = 0.8$. The lack of a phase shift between the tidal stress and the seismicity (Fig. 2) indicates that tidal loading must be under undrained conditions (see Supplementary Fig. 2 for a check on this assumption). Therefore, for calculations such as shown in Fig. 3b we use an effective friction $\mu' = (1 - B)\mu$, where $B$ is Skempton's coefficient. The proper value to use for $B$ is not well established: plausible values are between 0.5 and 1[17], so we explored values of $\mu'$ from 0.4 to 0. In Fig. 3b we used $\mu' = 0.4$, $K_m = 1$ GPa and $K_r = 55$ GPa.

The ratio $K_m/K_r$ and $\mu'$ control the $\Delta$CFS on overlying normal faults that results from an applied vertical tidal stress. We use a metric, $\chi$, which normalizes the $\Delta$CFS by the vertical tidal stress and which, therefore, is time invariant. We define $\chi$ by the $\Delta$CFS on a 67° dipping fault, indicated by the bold line in Fig. 3b, averaged from the corner of the magma chamber to the surface, normalized by the applied vertical tidal stress. This parameter is shown in Fig. 4 as a function of $K_m$ for several values of $\mu'$ at a fixed $K_r = 55$ GPa. Positive $\chi$ values indicate that normal faulting

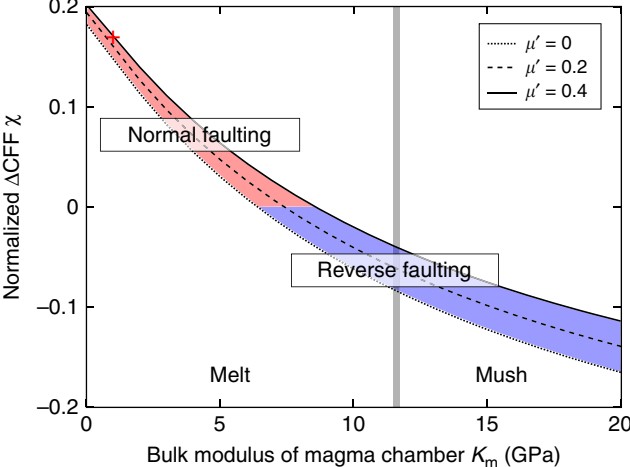

**Fig. 4** Properties of the magma chamber deformation system. The vertical axis $\chi$ is the average $\Delta$CFS on a 67° dipping normal fault from the tip of the magma chamber to the surface (bold line in Fig. 3b) resulting from an applied vertical tidal stress, normalized by that stress. The red area defines the conditions in which low tides encourage seismicity on normal faults and high tides discourage it, and the blue area vice versa. The plus sign indicates the conditions shown in Fig. 3b

earthquakes will be favored by low tides, negative values by high tides and vice versa for thrust earthquakes. All conditions in Fig. 4 within the red region therefore favor normal faulting earthquakes on the low tide and inhibit them on the high tide, and within the blue region, vice versa. Normal faulting may be generated by low tides for values of $K_m < 8$ GPa, depending on the value of $\mu'$. The point indicated by the plus in Fig. 4 is the case illustrated in Fig. 3b. The bulk modulus of gas-free mid-ocean ridge (MOR) magma is 12 GPa[18], but at the pressure of the magma chamber

(~40 MPa) this value can be reduced by one to two orders of magnitude by the presence of volatiles[19]. Thus, at this pressure, a magma of $K_m = 1$ GPa would contain 2650 ppm $CO_2$ by weight[18]. This is greater than the highest values typically seen for $CO_2$ content of MOR magma[20], but this difference could easily be accounted for by the inclusion of exsolved $H_2O$. So, the values of $K_m$ that we find would promote normal faulting at low tide are realistic. Our illustrative example (Fig. 3b) indicates $\chi = 0.176$, a figure that will enter into the modeling calculations of the triggered seismicity in the next section.

**Comparison of the earthquake triggering with theory**. During the three month period prior to the eruption, the seismic moment release on the eastern ring fault indicated a slip magnitude approximately the same as observed geodetically[12], indicating that the faults are seismically coupled. In this case, there are two models that relate change in seismicity rate to a rapid change in driving stress. These are based on earthquake nucleation models[21], one derived from the rate-state friction law[22] and the other from subcritical crack growth due to stress corrosion[23,24]. The rate-state friction version is

$$\frac{R}{r} = \exp\left[\frac{\Delta CFS}{A\sigma}\right], \tag{1}$$

and the stress corrosion version is

$$\frac{R}{r} = \left[1 + \left(\frac{\Delta CFS}{\Delta\tau}\right)^n\right], \tag{2}$$

where $R$ is the instantaneous seismicity rate, $r$ is the background rate, here taken as the rate when the tidal stress is zero, and $\Delta CFS = \chi\sigma_v$, $\sigma_v$ being the vertical tidal stress. The control parameters for the rate-state friction version are the effective normal stress $\sigma$ and the viscous friction term $A$. For the stress corrosion version, they are the stress corrosion index $n$ and the earthquake stress drop $\Delta\tau$.

The fit of these equations to the data is shown in Fig. 5, where the solid blue curve and the dashed red curves are Eqs. (1) and (2), respectively. These two formulations cannot be distinguished and fit the data equally well. There is no detectable phase shift between the seismicity and the tides (Fig. 2), nor is there any hysteresis observed—data for rising and falling stresses fit the triggering curves equally well (Supplementary Figs. 4 and 5). We, therefore, conclude that poroelastic relaxation is negligible in the response to the semi-diurnal tides.

The agreement with the theories is excellent, and extends them to far smaller stresses than previously seen[5], even into the negative stress regime. In the case shown in Fig. 5, the value of $\chi$ used was 0.176, from the illustrative example. The goodness of fit to the theories does not depend on the value of $\chi$ obtained from the deformation model: that merely determines the scale of the stress axis. The various implications of this result will be deferred to the discussion section.

**Applications to other areas**. Wilcock[6] searched for tidal triggering on the mid-ocean ridge systems of the NE Pacific, using mainly land-based networks. He found a 15% increase in seismicity within 15° of the lowest tides. The focal mechanisms of the earthquakes, however, were undetermined. With an OBS deployment on the Endeavour segment of the Juan de Fuca ridge, some 2° NE of Axial Volcano, the correlation of seismicity with low tides became much better defined[9]. Most of the triggered seismicity there was near the ridge axis, where the focal mechanisms indicate normal faulting[25]. This situation is therefore quite similar to Axial Volcano and the same effect of the magma chamber seems necessary to explain these observations.

At the hydrothermal field at 9°50′N on the East Pacific rise, an OBS deployment also showed evidence for tidal triggering[8]. There the ocean tides are much smaller than at Axial Volcano and a significant contribution to tidal stresses is from the solid earth tides. The seismicity maximum correlates with the maximum extensional tidal stress, which can reach 1.3 kPa. The dependence of the seismicity on stress is similar to that observed at Axial Seamount (compare Fig. 3c in ref. [8] to our Fig. 5). Evidence for the mechanism of the earthquakes is equivocal: scant focal mechanism data has indicated strike-slip, normal faulting, and reverse faulting[26,27], and others have proposed that the seismicity is due to hydrothermally induced extension cracking[28]. There is also a variation in the tidal phase angle of earthquakes along the strike of the ridge axis. This indicates the earthquake triggering is also modulated by pore pressure changes brought about by hydrothermal circulation[29]. With this degree of ambiguity and complexity, we cannot assess how deformation of the magma chamber may be related to the tidal triggering in this location.

The unloading model used here was initially tested at Katla volcano (Iceland), where earthquakes show an annual cycle with the maximum seismicity rate occurring in the late summer[30] when the snow cover of the glacier above the volcano is minimum (annual fluctuation—6 m). The model[31] showed that this was also the period of maximum Coulomb stresses in the area above the magma chamber. However, in this case the focal mechanisms were not known[30] so it was not possible to determine if the system was in the red or blue regions of Fig. 4.

**Discussion**
Our observations of seismicity rate change as a function of stress, shown in Fig. 5, show a much stronger agreement with the models and extends to much smaller stresses than previously observed[5]. In the rate-state friction version, the representative value $\chi = 0.176$ yields $A\sigma = 2$ kPa, about an order of magnitude smaller than found in earlier studies of earthquake triggering[5,32,33]. In the earlier studies the earthquakes were deeper (8–20 km), so the difference could be from that factor alone. In those papers, to accommodate lab values for $A$ of 0.003–0.007, near-lithostatic pore pressures were assumed to get low enough values of $\sigma$ to match the observed $A\sigma$. At Axial Volcano the normal stress is 7.2 MPa at the average earthquake depth of 1.2 km, assuming $\mu = 0.8$, a hydrostatic pore pressure gradient,

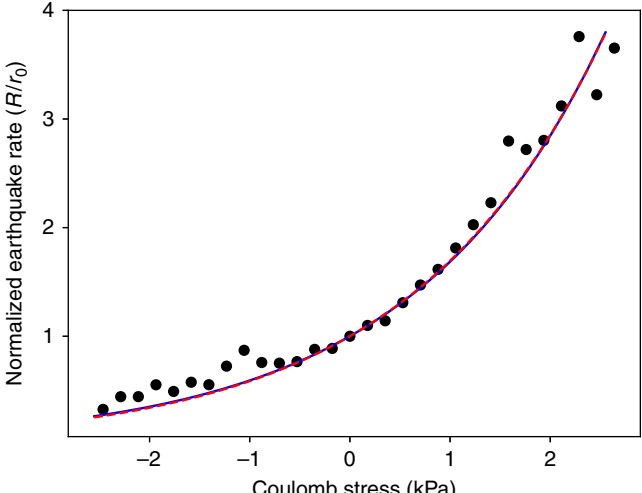

**Fig. 5** Normalized seismicity rate change vs. change in Coulomb stress. Coulomb stress is converted from tidal vertical stress using $\chi = 0.176$ (from the state indicated by the plus sign in Fig. 4). Blue curve is the rate-state friction version and the red curve is the stress corrosion version

and a dip of 67°. It is highly unlikely that overpressures can be maintained in the top 1 km of very young oceanic crust where there is no sediment cover and there is vigorous hydrothermal circulation throughout the caldera[34,35]. Using 7.2 MPa for $\sigma$ we conclude that $A = 0.0003$, much smaller than lab values. Considering the entire spread of the solution space for $\mu' \leq 0.4$, $\chi$ ranges from 0.2 to zero, and with $\sigma = 7.2$ MPa, the equivalent range of $A$ is $0.0004 \geq A > 0$. If hydrostatic pore pressure was assumed in the other studies, estimates for $A$ in that range would also be obtained. The higher sensitivity to triggering at Axial Volcano is due to the shallow depths of earthquakes there and its observation made clear by the high rate of background seismicity during the inflation stage. Considering the Vidale et al.[3] study, which failed to detect a tidal correlation for Southern California earthquakes, if we use 8 km depth with a hydrostatic pore pressure gradient, their typical tidal stress level of 1 kPa, and our $A$ value of 0.0003, we calculate[36] that they would need 22k events to detect a correlation, whereas their catalog contained only 13k. Thus, the other studies are consistent with our finding that the $A$ parameter at geologic rates must be considerably smaller than lab values. The few laboratory studies that explore if the friction rate parameters $A$ and $B$ depend on sliding rate[37,38] find that they do, and experiments at plate tectonic slip rates[39] indicate that the friction parameters at those rates may differ significantly from those measured at the much higher rates usually employed in laboratory experiments.

Beeler and Lockner[36] noted that there are two triggering regimes: a threshold regime, in which the earthquake nucleation time $t_n$ is shorter than the tidal period and a nucleation regime, in which it is longer. In the former, maximum seismicity rate would correlate with the maximum stressing rate, in the latter, with maximum stress amplitude. Our data clearly confirm the latter (Fig. 2), and the latter is also implicit in the fit in Fig. 5. The uplift rate prior to the 2015 eruption was 61 cm/y[13]. From our inflation model (e.g., Fig. 3a) we found that the corresponding fault stressing rate $\dot{\tau}$ is 5 MPa/y. Using[36] $t_n = \frac{2\pi A\sigma}{\dot{\tau}}$, we get $t_n = 24$ h confirming that the system is indeed in the nucleation regime for the semi-diurnal tides examined.

For the stress corrosion version of the triggering equation, if we take the stress corrosion index to be the laboratory values for basalt, $22 < n < 44$[40], then the best fitting stress drop would be $0.04 < \Delta\tau < 0.09$ MPa. This is a bit lower than the $0.18 < \Delta\tau < 2.8$ range[41] for earthquakes at 1 km depth in Southern California, although these estimates are from mainly strike-slip earthquakes, which have systematically higher stress drops than normal faults[42]. If we take the rule that stress drop is about 3% of the shear strength[42], then for strength $\tau = \mu\sigma = 5.7$ MPa we get $\Delta\tau = 0.17$ MPa, not much greater than our fit values. Thus, for this version of the triggering law, we do not have any serious conflict with independent estimates.

Thresholds for static or dynamic triggering have been much discussed[43–45]. Van der Elst and Brodsky[46] showed that dynamic triggering could be detected at very small strains, and suggested that the lower limit may simply be a matter of detectability. Our results (Fig. 5) show that seismicity rate falls smoothly as the tidal stress falls to zero, indicating that there is no threshold for triggering. Seismicity rate continues to fall when the tidal Coulomb stress becomes negative, indicating that what is often called stress shadowing is a continuous quantifiable function of stress reduction.

It has often been remarked that hydrothermal areas seem particularly susceptible to dynamic triggering from distant earthquakes[47–49] Attempts to explain this have invoked various effects of dynamic stresses on the permeability and/or pressure of the pore fluid[50–52]. The excellent agreement of our data with the dry triggering models indicates that additional mechanisms are not required to explain the tidal triggering at Axial Volcano. In the case of tidal triggering, some of those proposed mechanisms, such as unclogging of fluid pathways, are less likely because the tides are continually jostling the faults so that clogs, such as from mineralization as suggested in the Yellowstone case[48], will not have time to form.

Low-frequency earthquakes in the tremor deep in subduction zones also show high sensitivity to tidal stresses[53,54]. This undoubtedly implies very low effective normal stresses, although if our $A$ value is used instead of the laboratory values, effective stresses would be about an order of magnitude larger than those reported. This is more in line with the values of 2–4 MPa obtained from the scaling of the amplitude and frequency of the periodic slow slip events which the tremor accompanies[55].

The value of $A$ we obtained is the first estimate of a rate-state friction parameter for earthquakes at tectonic loading rates. As shown in the examples given above, using this value provides much more plausible estimates of effective normal stresses in those cases. This finding shows that laboratory values of friction parameters should not be blithely adopted in theoretical studies of earthquake phenomena.

## Methods

**Coulomb stress modeling.** Coulomb stress calculation is performed with the commercial Finite Element Modelling software COMSOL Multiphysics® (https://www.comsol.com). We use a $100 \times 100 \times 50$ km domain designed to limit boundary effects. Boundary conditions are zero-displacement for the bottom and lateral boundaries and free-displacement for the top boundary corresponding to the Earth's surface. For the host rock, we assume an isotropic and homogeneous elastic medium with a bulk modulus $K_r$ of 55 GPa and a Poisson's ratio $\nu_r$ of 0.25, which is in accordance with seismic velocities recorded on the East Pacific Rise[56]. At Axial Seamount, multichannel seismic-reflection has inferred a 14-km long by 3-km-wide shallow magma reservoir located at 1.1–2.3 km depth[15,16]. We, therefore, model the magma reservoir as a 3D ellipsoid cavity with semi-axis: $a = 7$ km, $b = 1.5$ km, and $c = 0.5$ km, and top depth located at 2 km below the surface.

In our modeling, the initial stress field is lithostatic and stress perturbations are calculated considering two scenarios: first, from the pressurization of the magma reservoir and second, for the effect of ocean tides. For the first scenario, the overpressure inside the reservoir is modeled by applying a constant normal stress applied at the boundary of the ellipsoid. For the second scenario, the stress changes due to ocean low tides are modeled by applying a boundary load at the surface corresponding to a 1 m decrease in the water level. Surface unloading causes the reservoir expansion resulting in a magma pressure change, which depends on the reservoir volume, the bulk modulus of the magma and the elastic properties of the host rock. The pressure change is applied on the reservoir's wall considering different bulk modulus $K_m$ from 0 to 20 GPa. For each model, the Coulomb failure stress change is calculated on specific fault planes using $\Delta CFS = \Delta\tau + \mu\Delta\sigma$, where $\Delta\sigma$ is the normal stress change, $\Delta\tau$ the tangential stress changes and $\mu$ the friction coefficient.

**Seismicity catalog.** The cabled seismic network started streaming time-corrected data in late January 2015, 3 months before Axial Volcano erupted on April 24th, 2015 (Wilcock et al., 2016)[7]. In this study, we examine the ~60,000 earthquakes located between January 22nd and April 23rd, 2015. The earthquake catalog is from Wilcock et al.[7] and is archived in the Interdisciplinary Earth Data Alliance Marine Geoscience Data System (DOI: 10.1594/IEDA/323843)[57] Tan et al.[14] estimated the $M_c$ of the catalog and find $M_c = 0.1$ when using the goodness-of-fit method[58] and $M_c = 0.3$ based on $b$-value stability[59].

**Tidal stress calculations.** The tidal stresses were modeled following Tan et al.[14] Briefly, we estimate the horizontal strains at the earthquake region (−130.009, 45.955) due to solid earth tide and regional ocean tidal loading with the SPOTL package[60] using the predicted tidal height in the Oregon State University regional ocean tidal model[61]. We then convert strain to stress using p-wave velocity of 5.55 km/s, s-wave velocity of 3.20 km/s, and a density of 2800 kg/m³, assuming plane stress. We also calculate the vertical stress variations due to near-field direct ocean tidal loading from the predicted tidal height taking seawater density of 1030 kg/m³ and gravitational acceleration of 9.8 m/s². We then estimate the horizontal stresses from the vertical stress assuming uniaxial strain. An example is shown in Supplementary Fig. 1.

The ratio between the horizontal and vertical tidal stresses is symmetrically distributed with a median value of 0.058 (Supplementary Fig. 4). Therefore, we use this ratio in the numerical modeling to obtain the $\chi$ value which relates the tidal vertical stress to the average Coulomb stress change on the normal fault (Fig. 4). This $\chi$ value (0.176 for $\mu' = 0.4$ and $K_m = 1$ GPa) is then used to convert the

tidal-vertical-stress time series to the Coulomb-stress time series for tidal triggering analysis (Fig. 5). This approximation gives almost identical results as when we use two separate $\chi$ values, $\chi_1 = 0.133$ relating the tidal vertical stress to the Coulomb stress change on the fault (with horizontal stress = 0) and $\chi_2 = 0.740$ relating the tidal horizontal stress to the Coulomb stress change on the fault (with vertical stress = 0), to convert the different component tidal-stress time series to the Coulomb-stress time series for tidal triggering analysis (Supplementary Fig. 5).

## Data availability

The earthquake catalog is from Wilcock et al.[7] and is archived in the Interdisciplinary Earth Data Alliance Marine Geoscience Data System (DOI: 10.1594/IEDA/323843).

## Code availability

Numerical models for Coulomb stress calculation have been performed using the software COMSOL Multiphysics® software [https://www.comsol.com] and source files are available from the co-author [F.A.] upon reasonable request.

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

## Acknowledgements

Mike Burton is thanked for instructing us on the properties of magmas at mid-ocean ridges. The earthquake catalog is the same as in Wilcock (2016) and is archived in the Interdisciplinary Earth Data Alliance Marine Geoscience Data System (DOI: 10.1594/IEDA/323843). That data was collected with funding from NSF under grant OCE-1536320. F. Albino is supported by the NERC Center for the Observation and Modelling of Earthquakes, Volcanoes and Tectonics.

## Author contributions

C.H.S. conceived the main ideas with discussions with Y.J.T., led the project, and wrote the initial draft of the paper. Y.J.T. performed the analysis of the tides and earthquake data, fit the triggering equations, contributed to their interpretation, and prepared Figs. 1, 2, and 5. F.A. developed the deformation model, carried out all deformation modeling, and prepared Figs. 3 and 4. All authors contributed to the writing of the final paper.

## Additional information

**Competing interests:** The authors declare no competing interests.

**Journal Peer Review Information:** *Nature Communications* thanks Elizabeth Cochran, Andrew Delorey and John Vidale for their contribution to the peer review of this work. Peer reviewer reports are available.

