## [Peer Review File · Nature Communications]

Reviewers' comments:

Reviewer #1 (Remarks to the Author):

I have read and considered this paper, but do not have a firm idea of whether its merits warrant publication in Nature.

The authors construct a model that can explain the observed correlation of earthquakes with ocean tides near mid-ocean ridges. Qualitatively, it makes sense, essentially by drawing faults that move to reverse the sense of correlation from a simpler structure, and the faults would be in the expected location. They also hypothesize that the magma chamber breathes in the expected way with the rise and fall of the tides overhead, although the fluctuations are greater than might be expected.

The value of this manuscript depend on two factors that are out of my purview. (1) Do the numbers make sense? I'm not a rate and state friction expert, so I don't know. Actually, few people are. (2) Is the structure plausible yet containing sufficient surprises that this is a valuable contribution? This is a question for people who understand the state of knowledge about fluid/solid magma chambers, and the fault systems near ridges. Again, not me.

I can offer no more definitive opinion than I see nothing major wrong with the arguments presented. Sorry not to be more decisive.

Reviewer #2 (Remarks to the Author):

The evidence and analysis provided in this manuscript are compelling and will be of interest to the geophysics community. I think an important result is the lower estimate for rate-and-state parameter A (viscous friction) and should be mentioned in the abstract.

Though the source of the earthquake catalog is cited, it would be helpful to the reader to provide some basic parameters such as number of earthquakes, duration of the catalog, and magnitude of completeness.

Not all cycles in the tidal function are the same, nor are they always symmetric. Based on the results presented, you state that there is a strong correlation between stress and seismicity rate that isn't sensitive to the stress path. In Figure S2, is the apparently higher rate during falling stresses, at the highest stresses, significant? It might be worth it to explore this a little more. For example, what if you plotted all the cycles where the initial stress is higher than the final stress by themselves and the same for the opposite case. Would they be different?

Lines 19, 126, 167

I admire the enthusiasm, but avoid using superlatives like "unprecedented".

Line 22

The first sentence does not adequately describe evidence for tidal triggering of continental earthquakes. Two studies are twenty and thirty years old; catalogs and analyses have evolved since then. The third looks at a single study area (onshore Japan) that can only marginally be called continental. Either do a more comprehensive review, or omit.

Line 181

Show the solution space you are considering on Figure 4. What is the valid range of the bulk modulus of the magma chamber to keep X positive? Since you are considering effective friction between 0 and 0.4 here, also show this on Figure 4.

Reviewer #3 (Remarks to the Author):

This manuscript examines the cause of tidal triggering at Axial Volcano located on the Juan de Fuca ridge. Clear tidal triggering at Axial Volcano during periods of inflation has been well documented by previous investigations, with increased seismicity occurring at low water levels. Recent work confirmed that tidally triggered seismicity is occurring on normal faults, which presents a problem for understanding why extensional vertical stresses would encourage failure on steeply dipping (67 degrees) normal faults. Vertical stresses applied to faults with dips greater than 45 degrees result in a larger component of shear stress change relative to normal stress change. Thus, steeply dipping normal faults should be promoted to fail at times of high water levels (compressional vertical stresses) when the shear stress promotes slip rather than low water levels as is observed. This work suggests that the existence of a highly compressible magma chamber (relative to surrounding rocks) beneath the normal faults may change the sign of the Coulomb stress change on the faults and explain the apparent paradox.

The paper could be significant, but I cannot evaluate the significance given what may be a critical error in the calculations. I believe there may be an error in the computation of Coulomb stress change. On Line 43 it is stated that tension is taken to be positive, but then on Line 64 the authors use the form of the Coulomb failure stress equation that is appropriate when compression is taken to be positive. The correct equation, if tension is positive, is $\Delta CFS = \Delta\tau + \mu\Delta\sigma$ (e.g. Harris, 1998; Stein, 1999). If this is not a simple typo limited to the text of the paper, then this has implications for the validity of all of the results of the paper.

Given this error, I cannot recommend the paper for publication at this time. If the authors are invited to revise their paper, I further hope they significantly revise the text and figures to improve clarity throughout. I found the paper difficult to follow given the imprecise word usage and lack of detail explaining the analyses undertaken.

Response to reviewer's comments

First of all, we thank reviewer 3 for pointing out the sign error in our equation for Coulomb stress. This resulted from a typo and has been corrected. All calculations of Coulomb stress have been checked and are correct.

Reviewer 1. This reviewer asks the question: Do these numbers make sense? We address this question in the Discussion section regarding our main finding, that the friction parameter A is much smaller than lab values. In particular, we add new text, lines 240-250, to show that this is an important finding that has implications throughout the area of theoretical studies of the earthquake mechanism. This reviewer also asks the question: Is the structure plausible yet containing sufficient surprises...?" The structure of the magma chamber itself is well mapped out by seismic imaging, refs. 15,16. What is new here is including the deformation of the magma chamber in determining the stress field induced by the tides.

Reviewer 2. This reviewer suggests that the lower estimate of the friction parameter A be mentioned in the abstracts. We agree and have rewritten the abstract to emphasize this aspect of our findings. As mentioned above, we have expanded the Discussion section to emphasize the importance of this finding.

We have now described the earthquake catalogue in more detail in the text (lines 51,52) and in much more detail now in the methods section (lines 504-511)

Re. the tidal stresses. We have now added an alternative and more precise method of calculating Coulomb stresses from tidal stresses in the Methods section (lines 513-533 and fig S2 and S3). This provides almost identical results as the more approximate method insofar as the fit to theory and the value of A obtained. With regards to the hysteresis test, he noted differences at the highest stresses between the rising and falling tide data. We think that this is because of poor sampling at those high stresses because of few data there. We have added a histogram of the stress distribution in Fig S5 to support this argument.

We have deleted the term 'unprecedented'

Line 22. We have expanded the historical review (lines 21-25). Ref 1 is a paper from 1897 showing negative results from the 19th Century. Ref 2 is a review paper showing negative results up to that time (1997) Ref 3 is a paper of 1998 showing negative results using modern methods. Ref 4 (2009) is the latest results that shows that one must use very large catalogues to detect a correlation for continental earthquakes.

Line 181, we now include a curve for $\mu=0$ so the entire solution space is delineated. We now state in the text that χ will be positive for $K_m < 8$ GPa.

Reviewer 3. The main concern of this reviewer, regarding the sign error in our Coulomb stress equation, has been addressed above.

This reviewer also wishes us to be more explicit in describing our methods of analysis. We have considerably expanded our Methods section to address that, and have improved the clarity of writing here and there as it seems fit. These places are highlighted in the text. Specific places include:

Lines 28-30, explaining Cochran's result

67-69 states that our model is a 3D model of the magma chamber

94-95 explaining better how we handle the poroelastic problem

104-106, adding material that shows that the faults are seismically coupled so that these models of triggering are appropriate (other triggering models have been developed for creeping faults)

136-140 explaining more carefully the dependence of the fit on parameters

240-245, adding a discussion of tidal triggering of tremor in subduction zones

Reviewers' comments:

Reviewer #2 (Remarks to the Author):

I have no additional comments after revisions. I support acceptance.

Reviewer #3 (Remarks to the Author):

Re-Review of:

The Mechanism of Tidal Triggering of Earthquakes at Mid-Ocean Ridges
Christopher H. Scholz, Yen Joe Tan, and Fabien Albino

This manuscript examines the cause of tidally modulated seismicity at Axial Volcano and infers that composition (e.g. bulk modulus) of the magma chamber is a critical factor in understanding the stress changes that trigger seismicity. A key observation is that no triggering threshold is observed (e.g. any stress change, no matter how small, impacts the likelihood of earthquake occurrence).

Thank you for checking and correcting the Coulomb stress equation – the modeling and conclusions are much easier to interpret now.

I suggest the manuscript is suitable for publication after some further minor to moderate clarifications that I delineate below.

Main comment:

I appreciate the effort the authors made to add more details to the methods. I suggest that additional detail is needed for the reader to understand how χ is determined. The current text states:

"The the [SIC] metric on the vertical axis, χ , is the Δ CFS on a 67° dipping fault averaged from the corner of the magma chamber to the surface, normalized by the vertical tidal stress. Positive χ values indicate that normal faulting earthquakes will be favored by low tides, negative values by high tides and vice versa for thrust earthquakes."

My notes are:

- How is exactly is χ computed? You have, for each grid point along your fault, a value for Δ CFS and a value for the vertical stress. So, are you taking the average of the ratios or the ratio of averages? Please clarify in the text your choice, and I wonder if an equation would help here.
- It is stated that χ is the average of values computed on a dipping fault from the corner of the magma chamber to the surface. It might be worth highlighting the relevant fault(s) in 3B – I believe I have guessed which fault(s) are being referred to but it is unclear.
- Given the discussion of Fig 3B (e.g. stress at 1 m low tide), I originally assumed χ was calculated for a certain vertical stress, e.g. at 1 m low tide. Additionally, the conditions modeled in Fig. 3B (at 1 m low tide) are called out by a plus sign on Fig. 4. But, the time varying vertical tidal stress results in a time varying CFS, and since χ is a ratio χ does not have to be tied to a specific (e.g. 1 m) tide. I wonder if this could be clarified for the reader to make it more obvious. For example, 'The bulk modulus of the magma chamber and the effective coefficient of friction control the resulting CFS on overlying normal faults in response to an applied vertical tidal stress. We use a metric, χ , that normalizes the CFS by the vertical tidal amplitude and, therefore, is time-invariant. Here, χ is calculated as...' [Or improved wording of your choice]

Other comments:

1. Line 94-95: Justify why undrained condition is appropriate here.
2. Line 98-99: What is meant by 'systematics of the system'? Please clarify.

3. Line 100: 'The the' typo.
4. Line 101: 'Corner of the magma chamber to the surface' is imprecise and should be clarified and/or the particular fault(s) highlighted in a figure.
5. Line 107-108: Consider changing to 'The plus sign in Fig. 4 indicates the case illustrated in Fig. 3b.'
6. Line 114: Change 'So values of Km within the red region are realistic.' to 'So, the values of Km that we find would promote normal faulting at low tide are realistic.'
7. Line 115: 'Our illustrative example indicates...' to 'Our illustrative example (Fig. 3b) indicates...'
8. Figure 1 is very sparse and you might consider adding tomography/seismic velocities as a basemap - I found some nice examples in other papers on Axial. And, a map showing us where Axial volcano is located should be included as Part A or as an inset of this figure. The two lines (sea floor? and magma chamber) are similar colors and are difficult to distinguish, and the sea floor line is not labeled or described. Does the figure use the highest resolution bathymetry available to show the sea floor?
9. Figure 3B. Highlight fault(s) used for χ calculation.
10. Figure 4. Δ CHS should be Δ CFS. In the caption, use the same terminology as the main text for the location of the fault, e.g. 'tip' versus 'corner' of the magma chamber. When the description of χ calculation is improved in the main text make sure those changes are reflected here. Y-axis could be 'Normalized Δ CFS, χ '. The symbol used to denote what is shown in figure 3B is a plus sign (+) not a cross (x); and correct this usage on Line 108.

Responses to comments of Reviewer #3

Main comment: Regarding how is χ computed... We have revised our text, lines 100-106, that specifically address the issues raised by the reviewer. A heavy line has been added to Fig 3B to indicate the fault where ΔCFS is calculated.

Minor comments

- 1 The reason for assuming undrained conditions is explained in lines 94-96. We also include an additional check on that assumption in figs S4 and S5. This additional test is also discussed in lines 140-143.
- 2 We removed this term 'systematics of the system' in that section and for the caption of fig 4, we call it 'properties of the system'.
- 3 Typo fixed
- 4 A bold line has been added to fig 3B to show this
- 5 We changed it accordingly
- 6 Lines 109-111 give the range that agrees with low tides stimulating seismicity. Line 117-118 has been changed as suggested
- 7 Line 118 changed as suggested
- 8 We have added a location inset to Fig. 1. We do not agree to adding velocity contours or other data as such will be extraneous clutter. We have added a description and new citation in the caption re. the bathymetry used.
- 9 Done
- 10 Captions for Figs 4 and 5 have been revised in accord with these comments.